# Hematological and biochemical alterations in preeclampsia: Readings from cord blood analysis

**Ahmed Abu Siniyeh[1]\*, Mohammad Alsahoury[2], Talal Al Qaisi[2,3¤], Majed Al-Holi[4]**

**1** Department of Medical Laboratory Sciences, School of Science, University of Jordan, Amman, Jordan, **2** Department of Medical Laboratory Sciences, Faculty of Allied Medical Sciences, Al-Ahliyya Amman University, Amman, Jordan, **3** Department of Biomedical Sciences, College of Health Sciences, Abu Dhabi University, Abu Dhabi, United Arab Emirates, **4** Cell Therapy Center, The University of Jordan, Amman, Jordan

¤ Current address: Department of Biomedical Sciences, College of Health Sciences, Abu Dhabi University, Abu Dhabi, United Arab Emirates
\* a.siniyeh@ju.edu.jo

## Abstract

### Background

Preeclampsia is a serious complication of pregnancy characterized by hypertension and proteinuria that adversely affects both maternal and fetal health. This study aimed to investigate hematological and biochemical alterations in cord blood associated with preeclampsia, with a focus on hemoglobin variants and blood gas parameters.

### Methods

A case–control study involving 54 participants, including 24 women diagnosed with preeclampsia and 30 normotensive controls, was conducted. Cord blood samples were analyzed for total hemoglobin (Hb), blood gas, and complete blood count (CBC) indices. Statistical analyses included independent t tests for parametric data and Mann–Whitney U tests for nonparametric data, with significance set at $p < 0.05$.

### Results

The results revealed significant differences in hemoglobin concentrations, with cord blood collected from preeclamptic women exhibiting lower levels of adult hemoglobin (HbA) ($64.0\% \pm 32.0\%$ vs. $76.2\% \pm 25.7\%$, $p = 0.004$) and higher fetal hemoglobin (HbF) concentrations ($35.9\% \pm 32.1\%$ vs. $23.7\% \pm 25.6\%$, $p = 0.004$) than controls. Blood gas parameters, including pH and bicarbonate and carbon dioxide levels, were not significantly different between the groups. However, CBC results revealed a lower platelet count in the cord blood of the preeclamptic group than in the cord blood of the preeclampsia group, ($213.7*10^3/\mu L \pm 112*10^3/\mu L$ vs. $314.6*10^3/\mu L \pm 70.8*10^3/\mu L$, $p = 0.0005$).

**Data availability statement:** The dataset generated and analyzed during the current study is available in Zenodo at https://doi.org/10.5281/zenodo.15294686.

**Funding:** The author(s) received no specific funding for this work.

**Competing interests:** The authors have declared that no competing interests exist.

## Conclusions

While our study reveals significant alterations in fetal hemoglobin variants and CBC indices in the cord blood of preeclamptic pregnancies, the clinical applicability of these markers for early detection is currently limited by the inaccessibility of fetal blood before delivery. Nevertheless, these findings offer important insights into the hematological changes linked to preeclampsia. Future studies should explore the potential of detecting similar alterations in maternal blood as a more feasible and non-invasive approach for early diagnosis and risk assessment of preeclampsia.

## Introduction

Preeclampsia is a clinical syndrome that occurs during the second trimester of pregnancy, affects 3–7% of pregnant women, and is one of the major causes of maternal morbidity and mortality, with a high prevalence in developing countries [1,2]. Preeclampsia is clinically defined as new-onset hypertension (systolic blood pressure ≥140 mmHg or diastolic ≥90 mmHg) after 20 weeks of gestation, accompanied by one or more of the following: proteinuria, maternal organ dysfunction (e.g., renal, hepatic, hematologic, or neurologic), or signs of uteroplacental insufficiency such as fetal growth restriction [3]. This condition is characterized by placental dysfunction, endothelial damage, and systemic inflammation, which can progress to severe complications such as HELLP syndrome, eclampsia, and preterm birth [4,5]. Despite its clinical importance, the pathophysiology of preeclampsia remains incompletely understood, and current management relies primarily on symptomatic treatment and timely delivery [6].

Hemoglobin (Hb) plays a key role in the proper transfer of oxygen from the mother to the fetus. In addition, it helps to remove carbon dioxide from the fetal blood back to the mother's blood [7]. During pregnancy, fetal hemoglobin (HbF) is the predominant oxygen carrier, exhibiting a higher affinity for oxygen than adult hemoglobin (HbA). This allows efficient oxygen extraction from maternal blood, which is essential for fetal growth and development [8]. Moreover, the level of HbA in fetal blood is low. This progressive HbA-related phenomenon reflects a transition from primarily HbF to progressively increasing HbA levels as pregnancy progresses [9].

Although the role of hemoglobin variants in fetal development is well established, their association with preeclampsia remains inadequately understood. The current literature provides limited and inconsistent data on changes in cord blood hemoglobin (HbF and HbA) and blood gas parameters such as oxygen [$O_2$] and carbon dioxide [$CO_2$] in preeclamptic pregnancies [10,11]. For example, some studies reported elevated hemoglobin levels in preeclampsia pregnancies, whereas others reported no significant differences compared with normotensive pregnancies [10,12]. Similarly, the relationships between preeclampsia and hematological indices, such as platelet counts and white blood cell (WBC) profiles, remain unclear, with conflicting findings across studies [13,14]. These gaps highlight the need for comprehensive investigations into the hematological and biochemical changes associated with preeclampsia.

This study aims to address these gaps by examining cord blood hemoglobin variants (HbF and HbA), blood gas parameters (O2 and CO2), and complete blood count (CBC) indices in preeclamptic and normotensive pregnancies. We hypothesize that preeclampsia is associated with distinct changes in cord blood hemoglobin profiles, blood gas parameters, and hematological indices compared with normotensive pregnancies. By identifying these differences, our findings may provide new insights into the pathophysiology of preeclampsia and contribute to the development of diagnostic and therapeutic strategies for this complex condition.

## Materials and methods

### Study design

This study was designed as a case–control investigation comprising two groups: those diagnosed with preeclampsia and those with normal pregnancies. The study was carried out from 19 May 2024 until 10 Dec 2024. This study aimed to analyze variations in cord blood hemoglobin, CBC indices, and gases between these two populations.

### Participants

The study involved a total of 54 participants, with 24 women with preeclampsia and 30 normal pregnant women.

### Ethics statement

Ethical approval was obtained from the **Ethics Committee for Scientific Research at the Ministry of Health, Jordan,** prior to the initiation of the study. The participants were thoroughly informed about the study's purpose and procedures, ensuring that their participation was voluntary. Confidentiality was strictly maintained; no identifying information, such as names, addresses, or national IDs, was disclosed. Written informed consent was obtained from all participants, emphasizing their right to withdraw from the study at any time without any impact on their medical care. As the study did not involve minors, parental or guardian consent was not needed. These measures were implemented to maintain the ethical integrity of the research and to protect the rights and welfare of all the participants involved.

### Inclusion and exclusion criteria

**Inclusion criteria.** Participants were divided into two groups. The preeclampsia group included women who were clinically diagnosed with preeclampsia based on the American College of Obstetricians and Gynecologists (ACOG) criteria, defined as new-onset hypertension (systolic blood pressure ≥140 mmHg and/or diastolic blood pressure ≥90 mmHg) accompanied by proteinuria (≥300 mg per 24 hours or ≥1+ on dipstick) after 20 weeks of gestation. The control group comprised normotensive pregnant women without any clinical or laboratory evidence of proteinuria, matched for gestational age.

**Exclusion criteria.** Exclusion criteria were clearly defined to reduce confounding variables. Women were excluded from the study if they had chronic hypertension prior to pregnancy, gestational diabetes mellitus (GDM), hematological disorders, renal or liver dysfunction, autoimmune or immunological conditions, fetal genetic abnormalities or structural anomalies, pregnancies achieved through in vitro fertilization (IVF), hormonal treatment during pregnancy, or a history of preterm labor.

### Data collection process

Medical histories and clinical data were retrieved through the Hakeem system, which is part of the National Electronic Health Record (EHR) infrastructure in Jordan. This system was used to access antenatal visit records, laboratory investigations, and other relevant patient information. Following delivery, umbilical cord blood samples were collected immediately and aseptically from each participant. The collection was done using two clamps on the umbilical cord. One clamp

was placed approximately 5 cm from the newborn's umbilical cord stump, whereas the second clamp was closer to the placenta. This method separated a portion of the umbilical cord and provides a sterile environment for blood extraction, thus preventing infection. This approach ensured accurate measurements of certain physiological parameters, since they are important in assessing maternal and fetal health, especially in normal pregnancy and preeclampsia.

### Arterial blood gas and electrolyte analysis

The umbilical arterial blood was assayed immediately after collection for pH, blood gases, and co-oximetry on the fully automated analyzer Roche Cobas b 123 POC System. Blood collection was performed directly into a syringe coated as an anticoagulant with dry, balanced heparin, which prevented clotting and thus ensured homogeneity in the sample. Blood samples were processed within 15 minutes after collection to ensure optimal data integrity. If this was not possible, samples were stored on-ice and processed within 30 minutes after extraction was essential for the integrity of data. Blood gases and acid-base parameters, including pH, partial pressures of carbon dioxide ($pCO_2$) and oxygen ($pO_2$), bicarbonate ($HCO_3$), base excess, and oxygen saturation ($SpO_2$), oxyhemoglobin ($O_2Hb$), carboxyhemoglobin ($COHb$), methemoglobin ($MetHb$), total bicarbonate ($cHCO_3$), and standard bicarbonate ($cHCO_{3-st}$) were measured. Electrolytes such as sodium ($Na^+$), potassium ($K^+$), chloride ($Cl^-$), and calcium ($Ca^+$) were also assessed.

**CBC analysis.** CBC testing was performed on the umbilical cord blood samples using a fully automated hematology analyzer (Sysmex XN-330), evaluating parameters including hemoglobin concentration, red and white blood cell counts, platelet count, and red cell indices including hematocrit value, mean corpuscular volume (MCV), mean corpuscular hemoglobin (MCH), and mean corpuscular hemoglobin concentration (MCHC).

**Hb Electrophoresis analysis.** The Hb Electrophoresis (hemoglobinopathy evaluation) test for the detection and analysis of Hb variants, HbA, HbF, and HbA2 fractions. This was done by completing the analysis with the use of the fully automated instrument H100 Hemoglobin Analyzer, based on principles from High-Performance Liquid Chromatography.

### Statistical analysis

Analyses were performed via Prism 8. Continuous variables (means ± SD) were assessed for normality Shapiro–Wilk test) and homogeneity of variance (Levene's test). Parametric data (e.g., tHb) were compared via independent *t* tests; nonparametric data (e.g., skewed distributions) were compared via Mann–Whitney *U* tests. A $p < 0.05$ indicated statistical significance.

## Results

### Demographic and health data

Analysis of demographic data comparing women with preeclampsia and those with normotensive pregnancies revealed several key differences. The data revealed that the mean age of women affected by preeclampsia was 31.90 years, in contrast to 27.30 years in the control group, which was a significant difference of 4.60 years (p = 0.017). Additionally, the mean gestational age seemed to be significantly lower in the preeclampsia group (35.80 weeks) than in the control group (38.60 weeks). Weekly averages were recorded, resulting in a significant difference (p < 0.001) of −2.80 weeks.

No statistically significant difference was found between the two groups in terms of educational attainment (p = 0.29). Similarly, although gestational weight was greater in the preeclampsia group (82.10 kg) than in the control group (76.00 kg), this difference did not reach statistical significance (p = 0.15). BMI was not significantly different between the groups (p = 0.98).

In addition, no meaningful differences were found in other lifestyle factors, such as smoking status and employment status (p = 0.35 and p = 0.88, respectively). Similarly, the menstrual duration and monthly salary were more similar between the two groups, with p values of 0.38 and 0.73, respectively. A trend indicating a significant difference in physician visit

frequency was observed, where the preeclampsia group had an average of 1.00 visits versus 0.83 in the control group (p = 0.06). In addition, the prevalence of previous cesarean deliveries and proteinuria was significantly greater in the preeclampsia group than in the control group (p < 0.001).

Regarding obstetric records, the preeclampsia subgroup had a greater number of previous pregnancies (3.30) than did the control group (1.80). However, this difference was not statistically significant (p = 0.08), nor were significant differences observed (Table 1).

## Cord blood gas results

This study evaluated several physiological and biochemical measures in the cord blood of a control group and a preeclampsia group. The mean pH for the preeclampsia group was 7.35 ± 0.09, whereas the control group had a mean of 7.37 ± 0.07, with no statistically significant difference (p = 0.540). The mean partial pressure of carbon dioxide ($pCO_2$) in the preeclampsia group was 45.16 ± 11.60, whereas that in the control group was 43.30 ± 7.00, indicating a non-significant difference (p = 0.469). The preeclampsia group had a mean partial pressure of oxygen ($pO_2$) of 22.91 ± 8.75, whereas the control group demonstrated a mean of 23.65 ± 6.04, with no statistically significant difference (p = 0.715). The preeclampsia group presented a mean hematocrit (Hct) of 38.5 ± 6.33, whereas the control group presented a mean of 39.8 ± 5.29, with no statistically significant difference (p = 0.612). The total hemoglobin (tHb) concentration in the preeclampsia group was 12.1 ± 1.67 g/dL, whereas that in the control group was 12.4 ± 1.55 g/dL, with no statistically significant difference.

**Table 1. Demographic and health data of the preeclampsia group vs the control group.**

| Variable | Preeclampsia n=24 (Mean±SD) | Control n=30 (Mean±SD) | p value |
|---|---|---|---|
| Age (years) | 31.9±7.63 | 27.3±5.75 | 0.017 |
| Gestational Age (weeks) | 35.8±2.85 | 38.60±2.42 | <0.001 |
| Education | 1.35±0.67 | 1.60±0.89 | 0.29 |
| Weight in Pregnancy (kg) | 82.10±14.05 | 76.00±16.31 | 0.15 |
| BMI | 25.42±4.46 | 25.39±5.01 | 0.98 |
| Smoking | 1.95±0.22 | 1.87±0.35 | 0.35 |
| Work Status | 1.15±0.37 | 1.17±0.38 | 0.88 |
| Menstrual Period Date (days) | 27.70±1.03 | 28.13±2.00 | 0.38 |
| Monthly Income | 1.35±0.49 | 1.40±0.50 | 0.73 |
| Doctor Visits | 1.00±0.00 | 0.83±0.38 | 0.06 |
| Previous Pregnancies | 3.30±0.94 | 1.80±0.31 | 0.08 |
| Multiple Pregnancies (Current) | 0.00±0.00 | 0.20±0.61 | 0.08 |
| Multiple Pregnancies (Previous) | 0.00±0.00 | 0.07±0.37 | 0.07 |
| Previous Miscarriages | 1.00±3.34 | 0.43±1.01 | 0.39 |
| Previous Cesareans | 0.75±1.07 | 0.00±0.00 | <0.001 |
| Chronic Disease | 2.00±0.00 | 2.00±0.00 | 1.00 |
| Diabetic (mmol/L) | 1.85±0.37 | 1.90±0.31 | 0.60 |
| Urinary Tract Infection | 1.45±0.51 | 1.57±0.50 | 0.43 |
| Proteinuria (mg/dL) | 2.00±0.00 | 1.05±0.22 | <0.001 |
| Male Newborns (%) | 60.9 | 56.7 | 0.72 |
| Female Newborns (%) | 39.1 | 43.3 | |

*Data are presented as the means±SDs, with significant differences noted (p < 0.05).

(p = 0.482). The SpO2 of the preeclampsia group was 46.40 ± 17.2, whereas that of the control group was 49.71 ± 16.2, with no statistically significant difference (p = 0.310).

Furthermore, the mean oxyhemoglobin (O$_2$Hb) level in the preeclampsia group was 45.28 ± 16.5, whereas that in the control group was 48.35 ± 15.4, with no statistically significant difference (p = 0.450). The level of carboxyhemoglobin (COHb) was equivalent in both groups, with a mean of 1.40, demonstrating no significant difference (p = 0.673). The mean methemoglobin (MetHb) level was 1.00 ± 0.23 in the preeclampsia group and 1.10 ± 0.09 in the control group, indicating no significant difference (p = 0.962).

The preeclampsia group presented a mean Deoxygenated hemoglobin (dHb) of 54.16 ± 17.3, whereas the control group presented a mean of 49.66 ± 16.2, with no statistically significant difference (p = 0.334). The total bicarbonate (cHCO$_3$) concentration in the preeclampsia group was 20.73 ± 2.89, whereas in the control group, it was 20.67 ± 1.95, indicating no significant difference (p = 0.933). The standard bicarbonate (cHCO$_{3-st}$) was not significantly different, with values of 19.58 ± 2.40 for the preeclampsia group and 19.81 ± 1.87 for the control group (p = 0.698).

The mean oxygen content in the preeclampsia group was 11.15 ± 3.06, whereas that in the control group was 12.39 ± 3.40, indicating no statistically significant difference (p = 0.251). The mean base excess in extracellular fluid (BE-EC) for the preeclampsia group was −5.850 ± 3.32, whereas in the control group, it was −5.534 ± 2.37, with no statistically significant difference (p = 0.683). The base excess (BE) was not significantly different between the groups (p = 0.683).

The results revealed no statistically significant differences in the analyzed physiological parameters between the preeclampsia and control groups, suggesting that preeclampsia conditions do not influence these specific blood gas and hemoglobin-related measures under the tested conditions (Table 2 and S1 Fig).

### Cord blood electrolytes results

Compared with those in the control group, the electrolyte levels in the preeclampsia group revealed the following results. For sodium (Na$^+$), the preeclampsia group had a mean of 133.1 ± 5.003, whereas the control group had a mean of 132.1 ± 2.42 (p = 0.389), which was not significant. The potassium (K$^+$) levels also did not significantly differ, with a mean of 5.615 ± 1.92 compared with 5.40 ± 1.29 in the control group (p = 0.890). Chloride (Cl$^-$) levels also did not significantly differ, with the preeclampsia group having a mean of 106 ± 3.55 compared with 106 ± 1.61 in the control group (p = 0.825). Moreover, the calcium (Ca$^+$) level was significantly different between the two groups, with a value of 1.11 ± 0.391, whereas it was 5.73 ± 23.8 in the control group (p = 0.0004) (Table 2).

### Cord blood complete blood count results

Table 3 shows the analysis of hematological parameters between women with preeclampsia and those with normal pregnancies, which revealed several significant differences. The analysis of the mean corpuscular hemoglobin concentration (MCHC) revealed no significant difference between the groups (p = 0.227), with the preeclampsia group having a mean of 32.36 ± 0.98 and the control group 32.68 ± 0.88. Similarly, the mean corpuscular hemoglobin (MCH) level was not significantly different (p = 0.169), with means of 35.55 ± 2.89 for the preeclampsia group and 34.62 ± 1.67 for the control group. However, Hb concentrations were significantly different, with the preeclampsia group averaging 15.01 ± 1.47 compared with 15.90 ± 1.44 in the control group (p = 0.029), indicating notable variance in this key measure. The analysis of RBC counts also revealed significant differences, with the preeclampsia group having a mean RBC count of 4.238 ± 0.42 and the control group having a mean RBC count of 4.601 ± 0.44 (p = 0.003). The mean packed cell volume (PCV) was 46.40 ± 4.46 in the preeclampsia group and 48.72 ± 4.82 in the control group, but this difference was not statistically significant (p = 0.075). The mean cell volume (MCV) was significantly different, as preeclampsia patients had a greater mean volume (110 ± 7.98) than did the control subjects (106 ± 4.76) (p = 0.043). Additionally, the red cell distribution width (RDW) was significantly different (p = 0.0425), with the preeclampsia group averaging 18.19 ± 2.38 and the control group averaging 17.02 ± 1.46.

**Table 2. Physiological and biochemical parameters in the cord blood of women with preeclampsia versus normal pregnant women.**

| Parameter | Preeclamp-sia n=24 (Mean±SD) | Control n=30 (Mean±SD) | p value |
|---|---|---|---|
| pH | 7.29±0.09 | 7.30±0.07 | 0.540 |
| $pCO_2$ (mmHg) | 45.16±11.60 | 43.30±7.00 | 0.469 |
| $pO_2$ (mmHg) | 22.91±8.75 | 23.65±6.04 | 0.715 |
| Hematocrit% | 38.5±6.33 | 39.8±5.29 | 0.612 |
| Total Hb (g/dL) | 15.8±1.67 | 16.7±1.55 | 0.482 |
| Oxygen saturation ($SpO_2$) % | 46.40±17.2 | 49.71±16.2 | 0.310 |
| Oxyhemoglobin ($O_2$Hb) % | 45.28±16.5 | 48.35±15.4 | 0.450 |
| Carboxyhemoglobin (COHb) % | 1.61±0.56 | 2.24±2.40 | 0.673 |
| Methemoglobin (MetHb) % | 1.06±0.23 | 1.07±0.09 | 0.962 |
| Deoxygenated Hb (dHb) % | 54.16±17.3 | 49.66±16.2 | 0.334 |
| Bicarbonate ($cHCO_3$) (mmol/L) | 20.73±2.89 | 20.67±1.95 | 0.933 |
| Standard bicarbonate ($cHCO_3$-st) (mmol/L) | 19.58±2.40 | 19.81±1.87 | 0.698 |
| Oxygen content % | 11.15±3.06 | 12.39±3.40 | 0.251 |
| Base excess in extracellular fluid (BE-ecf) (mmol/L) | −5.850±3.32 | −5.534±2.37 | 0.683 |
| Base excess (mmol/L) | −5.850±3.22 | −5.534±2.39 | 0.683 |
| $Na^+$ (mmol/L) | 133.1±5.003 | 132.1±2.42 | 0.389 |
| $K^+$ (mmol/L) | 5.615±1.92 | 5.40±1.29 | 0.890 |
| $Cl^-$ (mmol/L) | 106±3.55 | 106±1.61 | 0.825 |
| $Ca^+$ (mmol/L) | 1.11±0.391 | 5.73±23.8 | 0.0004 |

*Data are presented as the means±SDs, with significant differences noted (p<0.05).

**Table 3. CBC results of the preeclampsia group vs the control group.**

| Parameter | Preeclampsia n=24 (Mean±SD) | Control n=30 (Mean±SD) | p value |
|---|---|---|---|
| Mean corpuscular hemoglobin concentration (MCHC) | 32.36±0.98 | 32.68±0.88 | 0.227 |
| Mean corpuscular hemoglobin (MCH) | 35.55±2.89 | 34.62±1.67 | 0.169 |
| Mean Cell Volume (MCV) | 110±7.98 | 106±4.76 | 0.0431 |
| Hemoglobin (Hb) | 15.01±1.47 | 15.90±1.44 | 0.029 |
| RBC Count | 4.238±0.42 | 4.601±0.44 | 0.003 |
| Packed Cell Volume (PCV) | 46.40±4.46 | 48.72±4.82 | 0.075 |
| Red Cell Distribution Width (RDW) | 18.19±2.38 | 17.02±1.46 | 0.0425 |
| WBC Count | 10.15±5.21 | 13.81±3.71 | 0.006 |
| Neutrophil | 36.42±11.0 | 47.07±7.65 | 0.0001 |
| Lymphocyte | 51.46±11.8 | 39.87±8.32 | <0.0001 |
| Platelet | 213.7±112 | 314.6±70.8 | 0.0005 |

Moreover, the analysis of WBC counts revealed a significant difference between the groups (p=0.006), with the pre-eclampsia group averaging 10.15±5.21 and the control group averaging 13.81±3.71. This finding suggests a notable decrease in WBC counts in the preeclampsia group. The neutrophil count was highly significantly different and averaged 36.42±11.0 in the preeclampsia group compared with 47.07±7.65 in the control group (p=0.0001). Lymphocyte counts were also significantly higher in the preeclampsia group (51.46±11.8) than in the control group (39.87±8.32; p<0.0001).

Platelet counts also exhibited significant differences, with the preeclampsia group having a mean of 213.7±112 and the control group having a mean of 314.6±70.8 (p=0.0005), indicating a substantial decrease in platelet counts in the preeclampsia group.

F tests for variances associated with mean corpuscular hemoglobin, packed cell volume, and other parameters indicated no significant differences in variances, suggesting homogeneity across groups. Normality tests for residuals consistently showed that the assumptions of normality were met for all key measures, confirming the validity of our parametric tests.

In summary, our findings highlight significant differences in Hb and RBC counts, WBC counts, platelet counts, and red cell distribution width between the preeclampsia and control groups. Conversely, parameters such as the mean corpuscular hemoglobin concentration and packed cell volume did not significantly differ (Table 3 and S2 Fig).

### Hemoglobin electrophoresis results

The Mann–Whitney test was used to compare the concentration of HbF between the preeclampsia group and the control group. The analysis revealed a significant difference in HbF concentration, with a P value of 0.004 (exact), indicating a statistically significant difference (P<0.05). The preeclampsia group had a mean HbF concentration of 35.9±32.1, whereas the mean HbF concentration in the control group was 23.7±25.6, indicating a statistically significant increase, resulting in an actual difference of −12.2.

In contrast, HbA was significantly different between the preeclampsia group and the control group. The P value was 0.004 (exact), again indicating a statistically significant difference (P<0.05). The mean HbA concentration in the preeclampsia group was 64.0±32.0, whereas it was 76.2±25.7 in the control group.

Finally, the analysis of HbA2 did not reveal a significant difference between the two groups, with a P value of 0.082 (exact), suggesting no statistical significance (P>0.05). The mean HbA2 concentration in the preeclampsia group was 0.113±0.31, whereas it was 0.00±0.00 in the control group; however, this difference was not statistically significant (Table 4 and S3 Fig).

### Discussion

With a focus on hemoglobin-related metrics and blood gas concentrations in the umbilical cord blood from both women with preeclampsia and normotensive pregnancies, this study looked at several physiological findings. Preeclampsia may not impact these specific physiological variables under the examined conditions, since the results revealed no statistically significant variations in most parameters. Consistent with previous research, stable pH, $pCO_2$, bicarbonate ($cHCO_3$) and base excess (BE) concentrations indicate that preeclampsia does not cause major changes in acid–base balance or respiratory function, even though it is linked to various physiological changes [15].

The stability of the acid–base balance indicates that women with preeclampsia maintain normal physiological function, which is consistent with the observations of Brichant et al. (2010), who reported maintaining acid–base homeostasis in the cord blood of women with preeclampsia. Moreover, the $pCO_2$ exhibited negligible fluctuations, suggesting that respiratory

**Table 4. Hb electrophoresis results of the preeclampsia group vs the control group.**

| Parameter | Preeclampsia n=24 (Mean±SD) | Control n=30 (Mean±SD) | P Value |
|---|---|---|---|
| HbF | 35.9±32.1 | 23.7±25.6 | 0.004 |
| HbA | 64.0±32.0 | 76.2±25.7 | 0.004 |
| HbA2 | 0.113±0.31 | 0.00±0.00 | 0.082 |

function is preserved in this cohort [16]. On the other hand, a study highlighted that while overall base excess (BE) in severe preeclampsia is similar to that in healthy pregnancies, preeclampsia is associated with greater imbalances due to hypoalbuminemic alkalosis and hyperchloremic acidosis. This finding suggests that there is significant acid–base disturbances do exist in women with preeclampsia, which may not be reflected in stable pH and $pCO_2$ values alone [17]

Oxygen saturation ($SpO_2$) and oxyhemoglobin ($O_2Hb$) concentrations in the cord blood samples were comparable across the groups, indicating that there was no significant reduction in $O_2$ transport or availability in patients with preeclampsia [18]. On the other hand, a study using near-infrared spectroscopy has shown that placental tissue oxygen saturation ($SpO_2$) is significantly lower in complicated pregnancies, such as those with preeclampsia, than in uncomplicated pregnancies [19]. This reduced saturation is indicative of compromised placental function, which can lead to inadequate fetal oxygenation. While oxyhemoglobin ($O_2Hb$) saturation may appear normal, the underlying vascular and metabolic changes in preeclampsia can significantly compromise effective oxygen transport, highlighting the need for comprehensive assessments beyond standard oxygen saturation metrics.

No significant differences in cord blood sodium ($Na^+$), potassium ($K^+$), or chloride ($Cl^-$) levels were detected between preeclamptic and normotensive pregnancies, which aligns with prior reports that these electrolytes remain stable despite placental dysfunction [20,21]. However, a striking reduction in cord blood calcium ($Ca^+$) levels was observed in the preeclampsia group, which is consistent with growing evidence linking hypocalcemia to preeclampsia pathogenesis [22,23]. This disparity may reflect altered placental calcium transport mechanisms, as maternal–fetal calcium transfer is critical for fetal homeostasis and is disrupted in preeclampsia due to impaired trophoblast function and oxidative stress [22]. The marked variability in Ca levels (SD ± 23.8) in the control group suggests potential methodological or sampling heterogeneity, warranting cautious interpretation.

Additionally, assessments of hematocrit (Hct), total hemoglobin (tHb), and other Hb derivatives, such as methemoglobin (MetHb) and carboxyhemoglobin (COHb) in the cord blood, revealed no significant changes. These findings suggest that preeclampsia does not markedly influence the total blood volume or Hb concentration, thus confirming the stability of the oxygen-carrying capacity in these patients [10]. On the other hand, an earlier study revealed that in umbilical cord blood from newborns of preeclamptic mothers, there were significant increases in RBC count, hemoglobin, hematocrit, and mean cell hemoglobin concentration [24]. The results of the study by Catarino et al. are also inconsistent with our findings that showed decreased RBC counts and Hb concentrations in the cord blood of women with preeclampsia.

While the WBC count was low, in cord blood of preeclamptic women, the lymphocyte count was elevated, and the neutrophil count was decreased. Sultana et al. (2022) reported that the total WBC count and absolute neutrophil count in the cord blood of newborns from preeclamptic mothers were significantly lower than those in the cord blood of healthy mothers, but lymphocyte counts were not specifically reported [25]. In contrast, WBC and neutrophil percentages were found to be elevated, whereas the lymphocyte percentage was decreased in preeclamptic patients, indicating a shift in the adaptive immune response to the inflammatory state associated with preeclampsia [26,27].

The notable shift in platelet counts observed in the cord blood of preeclampsia group represents another important finding, as the preeclampsia group presented a significantly lower platelet count. Endothelial injury and microvascular thrombosis are the main causes of platelet activation and aggregation, which is why thrombocytopenia is common in severe preeclampsia [28]. Monitoring platelet counts in women with preeclampsia is crucial for evaluating disease progression and anticipating complications such as HELLP syndrome. The findings of this study align with those of previous studies linking thrombocytopenia to the severity of preeclampsia [29–31].

Our findings suggest that alterations in fetal hemoglobin profiles namely, increased levels of fetal hemoglobin (HbF) and decreased levels of adult hemoglobin (HbA) may reflect impaired oxygen exchange and placental insufficiency, both of which are central features of preeclampsia. While these hemoglobin variants were measured in cord blood and are therefore not directly applicable for early screening, they may serve as indicators of systemic changes that could also manifest in maternal blood. Previous studies have suggested that elevated HbF in maternal circulation may occur

due to placental leakage and could serve as an early biomarker of preeclampsia risk [32,33]. In parallel, hematological parameters such as low platelet counts and elevated neutrophil–to-lymphocyte ratios observed in our preeclampsia group have also been reported as potential markers of inflammation and vascular dysfunction in preeclamptic pregnancies [34]. These results support the possibility that a panel of maternal blood tests incorporating hemoglobin fractions and CBC indices could be developed for risk assessment and monitoring. Further studies are warranted to validate these findings in larger cohorts and to investigate their translational potential for early detection in routine prenatal care.

The cord blood of preeclamptic women had significantly lower HbA concentrations, which may indicate an abnormality in the normal process of Hb formation. In brief, the reduction in HbA and the absence of HbA2 could indicate potential issues in erythropoiesis caused by chronic hypoxia and oxidative stress in preeclampsia [35]. In preeclampsia-affected pregnancies, the decreased concentration of HbA and increased concentration of HbF in cord blood suggests a complex interplay between placental injury, oxidative stress, and altered blood function [36].

Our findings are consistent with those of Vijaya et al. (2018), who reported a strong correlation between HbF and the onset of preeclampsia. Additionally, in preeclamptic women, HbF may play a role in the development of eclampsia [37]. Preeclampsia is characterized by oxidative stress evident in both maternal circulation and the placenta, with a reduced placental antioxidant capacity compared with that in normal pregnancies. The condition is theorized to originate in the placenta, culminating in maternal endothelial dysfunction, driven in part by oxidative stress from free fetal hemoglobin (HbF), which accumulates in preeclamptic placentas, as shown by gene and protein studies. This oxidative stress initiates a pathogenic cycle in the early stages, exacerbating clinical symptoms later through excessive hemoglobin release—HbF in preeclampsia and HbA in HELLP syndrome. Maternal constitutional factors influence susceptibility to oxidative stress, modulating disease severity and manifestations [38]

## Limitations

In the context of Jordan, the overall incidence rate of preeclampsia, which is considered a rare disease, is reported to be 1.3% [39]. The small sample size in this study, which was due to the low incidence of preeclampsia and difficulties in sample collection, may limit the generalizability of our findings. The sample size limitation could influence the statistical ability to detect differences in such parameters. Therefore, further research with larger sample sizes is needed to validate these findings and provide more robust conclusions.

Another limitation of this study is the significant difference in gestational age between the preeclampsia and control groups, with earlier deliveries observed in the preeclampsia group due to clinical management decisions. Since hematological and biochemical parameters may vary with gestational age, the observed differences in cord blood profiles could partly reflect gestational age effects. Future research should consider gestational age matching or apply statistical adjustments to account for this potential confounder.

## Conclusions

Although our study demonstrates significant alterations in fetal hemoglobin variants (HbF and HbA) and CBC indices in the cord blood of preeclamptic pregnancies, we acknowledge that the direct use of these fetal markers for early detection in routine clinical practice may be limited due to the inaccessibility of fetal blood prior to delivery. However, these findings may provide valuable pathophysiological insights into the hematological changes associated with preeclampsia. Future research should investigate whether similar alterations can be detected in maternal blood, which is more readily accessible during pregnancy. Identifying maternal biomarkers that reflect fetal or placental stress, such as free HbF or shifts in platelet indices, could potentially serve as non-invasive tools for earlier diagnosis and monitoring of preeclampsia risk. While our findings provide valuable insights into the immune changes related to preeclampsia, it should be kept in mind that preeclampsia is by no means caused by a single factor or single cause of disease. Genetic predispositions and

environmental influences may strongly influence the progression and development of this disease. Hence, further studies are certainly needed to explore these factors across larger and more diverse populations.

## Supporting information

**S1 Dataset. Raw data generated during the current study.** The dataset used for analysis in this study is available at Zenodo: https://doi.org/10.5281/zenodo.15294686.
(XLSX)

**S2 Dataset. Analyzed data from the current study.** The dataset containing processed/summary data used in the analyses is available at Zenodo: https://doi.org/10.5281/zenodo.15294686.
(XLSX)

**S1 Fig. Comparison of cord blood gas parameters between control and preeclampsia groups.** The graphs present various parameters of cord blood gas in control group vs preeclampsia group. The graphs illustrate the following **(A)** pH levels **(B)** Partial pressure of carbon dioxide ($pCO_2$). **(C)** Partial pressure of oxygen ($pO_2$). **(D)** Oxygen saturation ($SpO_2$). **(E)** Oxyhemoglobin ($O_2Hb$). **(F)** Carboxyhemoglobin (COHb). **(G)** Methemoglobin (MetHb). **(H)** Deoxygenated hemoglobin (dHb). **(I)** Concentration of bicarbonate ($cHCO_3$). **(J)** Standardized bicarbonate ($cHCO_3$-st). **(K)** Total oxygen content ($ctO_2$). **(L)** Base excess in extracellular fluid (BE-ecf). **(M)** Base excess (BE). **(N)** Hematocrit (Hct). **(O)** Total hemoglobin (tHb). Significant findings are indicated by asterisks (*) for statistically significant differences, while "ns" denotes non-significant results.
(TIF)

**S2 Fig. Comparison of complete blood count parameters between control and preeclampsia groups.** This figure displays various complete blood count (CBC) parameters measured in a control group and preeclampsia group. Panels (A) through (K) illustrate the following hematological parameters: **(A)** hemoglobin (Hb) levels, **(B)** packed cell volume (PCV), **(C)** red blood cell (RBC) count, **(D)** mean corpuscular volume (MCV), **(E)** red cell distribution width (RDW), **(F)** mean corpuscular hemoglobin (MCH), **(G)** mean corpuscular hemoglobin concentration (MCHC), **(H)** platelet count, **(I)** white blood cell (WBC) count, **(J)** neutrophil count, and **(K)** lymphocyte count. Significant findings are indicated by asterisks (*) for statistically significant differences, while "ns" denotes non-significant results.
(TIF)

**S3 Fig. Hemoglobin electrophoresis comparison between control and preeclampsia groups.** Hemoglobin electrophoresis analysis comparing the preeclampsia and Control groups. The concentration of HbA and HbF is significantly different between the two groups, while $HbA_2$ shows no significant difference.
(TIF)

## Acknowledgments

We would like to thank Al Hussein Hospital, the doctors and nursing staff in the Obstetrics and Gynecology Department for their efforts in collecting samples.

## Author contributions

**Conceptualization:** Majed Al-Holi.

**Data curation:** Ahmed Abu Siniyeh.

**Formal analysis:** Ahmed Abu Siniyeh.

**Investigation:** Mohammad Alsahoury, Majed Al-Holi.

**Methodology:** Mohammad Alsahoury.

**Project administration:** Ahmed Abu Siniyeh, Talal Al Qaisi.

**Supervision:** Ahmed Abu Siniyeh, Talal Al Qaisi.

**Validation:** Talal Al Qaisi.

**Visualization:** Mohammad Alsahoury, Talal Al Qaisi.

**Writing – original draft:** Ahmed Abu Siniyeh.

**Writing – review & editing:** Ahmed Abu Siniyeh, Talal Al Qaisi, Majed Al-Holi.

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
