## [Decision Letter · Decision Letter 0]

8 Apr 2025

PONE-D-25-07466Hematological and Biochemical Alterations in Preeclampsia: Readings from Cord Blood AnalysisPLOS ONE

Dear Dr. Siniyeh,

Thank you for submitting your manuscript to PLOS ONE. After careful consideration, we feel that it has merit but does not fully meet PLOS ONE’s publication criteria as it currently stands. Therefore, we invite you to submit a revised version of the manuscript that addresses the points raised during the review process.

We look forward to receiving your revised manuscript.

Kind regards,

Apeksha Niraula, M.D.

Academic Editor

PLOS ONE

Journal Requirements:

2. In the online submission form you indicate that your data is not available for proprietary reasons and have provided a contact point for accessing this data. Please note that your current contact point is a co-author on this manuscript. According to our Data Policy, the contact point must not be an author on the manuscript and must be an institutional contact, ideally not an individual. Please revise your data statement to a non-author institutional point of contact, such as a data access or ethics committee, and send this to us via return email. Please also include contact information for the third party organization, and please include the full citation of where the data can be found.

Additional Editor Comments:

1. Language editing is must for the manuscript.

2. Author highlighted the potential for using hemoglobin variants and CBC indices as biomarkers for early detection and management of the condition, but has not described about it.

3. Methodology needs to be elaborated and clear. Inclusion criteria must be properly illustrated.

4. Results and Discussion needs to be reformatted and rewritten.

Reviewers' comments:

Reviewer's Responses to Questions

**Comments to the Author**

1. Is the manuscript technically sound, and do the data support the conclusions?

Reviewer #1: Yes

Reviewer #2: Partly

2. Has the statistical analysis been performed appropriately and rigorously? 

Reviewer #1: Yes

Reviewer #2: Yes

3. Have the authors made all data underlying the findings in their manuscript fully available?

Reviewer #1: Yes

Reviewer #2: Yes

4. Is the manuscript presented in an intelligible fashion and written in standard English?

Reviewer #1: No

Reviewer #2: Yes

5. Review Comments to the Author

Reviewer #1: There are many sentences that do not correspond to academic and formal English writing. The quality of presentation needs to be improved.

For example, the following sentence looks good at first reading, but there is an editing error when considered in detail:

Analysis of demographic data comparing preeclampsia women and the control group displayed several key differences.

Instead of calling one group the women and the other the control group, a simpler fade and a sentence like Analysis of demographic data of the study groups (or of women with or without preeclampsia) revealed some differences may suffice.

Reviewer #2: The authors examined whether preeclampsia alters cord blood haematology and biochemistry. The manuscript shows that adult haemoglobin is reduced and foetal haemoglobin is increased in the cord blood of preeclampsia. I have some comments on the manuscript below:

Abstract:

Results, Lines 6-7 - Please remove ‘with lower platelet counts’ from the following statement as the direction of change has already been mentioned: ‘However, CBC results indicated lower platelets count in the cord blood of preeclamptic group, with lower platelet counts’.

Conclusions, Lines 3-5 - The authors state that ‘…highlight the potential for using haemoglobin variants and CBC indices as biomarkers for early detection and management of the condition’. Would this be possible in routine clinical practice as the haemoglobin variants measured were in foetal blood and not maternal blood.

Background:

Paragraph 1, Line 4 - The current diagnostic criteria for preeclampsia is the de novo onset of high blood pressure (systolic ≥140 mmHg or diastolic ≥90 mmHg) after 20 weeks of gestation, along with either proteinuria, signs of organ dysfunction, and/or uteroplacental insufficiency.

Methods:

Inclusion criteria – the authors state that the preeclamptic group included women who has been previously diagnosed with preeclampsia. Do the authors mean these women previously had a pregnancy complicated by preeclampsia or that their current pregnancy is complicated with preeclampsia. It would be useful to know what the criteria for preeclampsia was for the study.

Results:

As the results currently are presented, the data are presented as a Table with many parameters also included in a Figure format. Please update the presentation of results to present that data in only one format (i.e., Table or Figure).

Demographic and health data, Paragraph 1, Lines 4-7 – Was the reduction in gestational age at birth in the preeclampsia group due to the need to prematurely deliver the baby for maternal and foetal health outcomes? If so, was the accounted for with similar gestational age-matched samples from uncomplicated pregnancies?

It would be useful to know what proportion of women underwent vaginal vs. caesarean delivery.

Table 1 – It would be useful to know what the units are for some of the variables included. For example, proteinuria and diabetic?

Additionally, what is the proportion of male and female babies in each group?

Please change the following title ‘Cord blood electrolytes results’.

Please change the following title ‘Maternal complete blood count results’. The ‘Data Collection

Process’ only states that CBC was performed in maternal blood. If it was performed on cord blood, please include specific details of the exact analysis performed in the analysis to improve the clarity of the study as the current list is quite vague for the ‘various parameters’ measured.

Discussion:

The wording in the manuscript could be improved to specifically state whether the parameters being discussed (i.e., from the study vs. current literature) are in maternal or cord blood. It is often difficult to know whether interpretations/statements made are in reference to maternal or foetal outcomes and how the link to what is known in the literature.

Conclusion - As the authors only investigated cord blood at term, it is difficult to know whether HbF and HbA could be used are biomarkers for early detection and management without knowing whether these parameters are changes prior to disease onset.

6. PLOS authors have the option to publish the peer review history of their article (what does this mean? ). If published, this will include your full peer review and any attached files.

**Do you want your identity to be public for this peer review?** For information about this choice, including consent withdrawal, please see our Privacy Policy .

Reviewer #1: **Yes: ** Ali Cetin

Reviewer #2: No

---

## [Author Response · Author response to Decision Letter 1]

22 Apr 2025

'Response to Reviewers'

Journal Requirements:

Author response:

Dear Editor,

We have reviewed the manuscript and confirm that all PLOS ONE style requirements, including those related to authorship and reference style, have been fully addressed.

Please let us know if any further modifications are needed.

2. In the online submission form you indicate that your data is not available for proprietary reasons and have provided a contact point for accessing this data. Please note that your current contact point is a co-author on this manuscript. According to our Data Policy, the contact point must not be an author on the manuscript and must be an institutional contact, ideally not an individual. Please revise your data statement to a non-author institutional point of contact, such as a data access or ethics committee, and send this to us via return email. Please also include contact information for the third party organization, and please include the full citation of where the data can be found.

Author response:

Dear Editor,

Thank you for your feedback regarding the Data Availability Statement.

We would like to clarify that all data underlying the findings of our study are fully available without restriction. The previous reference to the corresponding author as the data access contact was unintentional and we appreciate your guidance. We would like to clarify that all relevant data underlying the findings of our manuscript are fully available and included within the manuscript and its Supporting Information files.

Accordingly, please find below our revised Data Availability Statement:

All relevant data are within the manuscript and its Supporting Information files.

We appreciate your guidance and look forward to the next steps in the review process. Please let us know if any additional information is required.

Ahmed Abu Siniyeh

Author response:

Dear Editor,

Thank you for your comment regarding the ethics statement. We confirm that ethical approval for this study was obtained, and we have now included the full ethics statement in the Materials and Methods section of the revised manuscript as follows:

“Ethics statement

Ethical approval was obtained from the Ethics Committee for Scientific Research at the Ministry of Health, Jordan, prior to the initiation of the study. The participants were thoroughly informed about the study's purpose and procedures, ensuring that their participation was voluntary. Confidentiality was strictly maintained; no identifying information, such as names, addresses, or national IDs, was disclosed. Written informed consent was obtained from all participants, emphasizing their right to withdraw from the study at any time without any impact on their medical care. As the study did not involve minors, parental or guardian consent was not needed. These measures were implemented to maintain the ethical integrity of the research and to protect the rights and welfare of all the participants involved.”

The paragraph is highlighted in RED in “Manuscript” file. You can also check it in the “Revised Manuscript with Track Changes” file

Please let us know if any additional details or revisions are needed. We appreciate your guidance and continued consideration of our manuscript.

Additional Editor Comments:

1. Language editing is must for the manuscript.

Author response: You are correct. We have carefully revised the manuscript for grammar, clarity, and overall flow. A professional language editing service was also consulted to ensure that the manuscript now meets academic and journal standards for English language quality.

Please check “Revised Manuscript with Track Changes” file.

2. Author highlighted the potential for using hemoglobin variants and CBC indices as biomarkers for early detection and management of the condition, but has not described about it.

Author response: Thank you for this important observation. We have revised and expanded the Discussion section of the manuscript to more clearly describe the potential clinical relevance of hemoglobin variants (HbF and HbA) and CBC indices (such as platelet count, neutrophil and lymphocyte percentages) as early biomarkers in preeclampsia.

In particular, we now explain that elevated fetal hemoglobin (HbF) and reduced adult hemoglobin (HbA) levels in cord blood may reflect chronic fetal hypoxia and placental dysfunction—hallmarks of preeclampsia. These changes, while currently identified in fetal samples, may suggest underlying processes that could also be detectable in maternal blood. We emphasize the need for further research to explore whether these hemoglobin patterns, or their maternal correlates, could be adapted for non-invasive early screening tools.

Additionally, we highlight the clinical significance of hematological indices such as low platelet counts and shifts in leukocyte subtypes, which have previously been associated with inflammatory states and endothelial dysfunction in preeclampsia. For example, increased neutrophil-to-lymphocyte ratios and thrombocytopenia may serve as potential early warning indicators of disease onset or severity.

We have also cited relevant literature to support these points, including studies by Anderson et al. (2011), which investigated HbF as a predictive marker for preeclampsia, and Kirbas et al. (2015), who examined CBC parameters as part of early risk assessment.

Furthermore, to acknowledge the limitations of our findings in fetal (cord blood) samples, we now clarify that while these results may not yet be applicable for direct clinical use, they provide a foundation for developing future maternal biomarkers that reflect similar physiological disruptions.

We appreciate your comment, which helped us improve the clarity and clinical contextualization of our findings.

Here, the paragraph added to the Discussion section and highlighted in RED in the revised manuscript

“Our findings suggest that alterations in fetal hemoglobin profiles namely, increased levels of fetal hemoglobin (HbF) and decreased levels of adult hemoglobin (HbA) may reflect impaired oxygen exchange and placental insufficiency, both of which are central features of preeclampsia. While these hemoglobin variants were measured in cord blood and are therefore not directly applicable for early screening, they may serve as indicators of systemic changes that could also manifest in maternal blood. Previous studies have suggested that elevated HbF in maternal circulation may occur due to placental leakage and could serve as an early biomarker of preeclampsia risk [33, 34]. In parallel, hematological parameters such as low platelet counts and elevated neutrophil-to-lymphocyte ratios observed in our preeclampsia group have also been reported as potential markers of inflammation and vascular dysfunction in preeclamptic pregnancies [35]. These results support the possibility that a panel of maternal blood tests incorporating hemoglobin fractions and CBC indices could be developed for risk assessment and monitoring. Further studies are warranted to validate these findings in larger cohorts and to investigate their translational potential for early detection in routine prenatal care.”

3. Methodology needs to be elaborated and clear. Inclusion criteria must be properly illustrated.

Author response: Thank you for this valuable feedback. We have revised and expanded the Materials and Methods section to ensure clarity and transparency in study design, participant selection, and data collection. The following specific changes have been made and The paragraph is highlighted in RED in “Manuscript” file. You can also check it in the “Revised Manuscript with Track Changes” file

“Inclusion and Exclusion Criteria

Inclusion criteria:

Participants were divided into two groups. The preeclampsia group included women who were clinically diagnosed with preeclampsia based on the American College of Obstetricians and Gynecologists (ACOG) criteria, defined as new-onset hypertension (systolic blood pressure ≥140 mmHg and/or diastolic blood pressure ≥90 mmHg) accompanied by proteinuria (≥300 mg per 24 hours or ≥1+ on dipstick) after 20 weeks of gestation. The control group comprised normotensive pregnant women without any clinical or laboratory evidence of proteinuria, matched for gestational age.

Exclusion criteria:

Exclusion criteria were clearly defined to reduce confounding variables. Women were excluded from the study if they had chronic hypertension prior to pregnancy, gestational diabetes mellitus (GDM), hematological disorders, renal or liver dysfunction, autoimmune or immunological conditions, fetal genetic abnormalities or structural anomalies, pregnancies achieved through in vitro fertilization (IVF), hormonal treatment during pregnancy, or a history of preterm labor.

Data collection process

Medical histories and clinical data were retrieved through the Hakeem system, which is part of the National Electronic Health Record (EHR) infrastructure in Jordan. This system was used to access antenatal visit records, laboratory investigations, and other relevant patient information. Following delivery, umbilical cord blood samples were collected immediately and aseptically from each participant. The collection was done using two clamps on the umbilical cord. One clamp was placed about 5 cm from the newborn's stump of the umbilical cord, whereas the second clamp was closer to the placenta. This method separated a portion of the umbilical cord and provides a sterile environment for blood extraction, thus preventing infection. This approach ensured accurate measurements of certain physiological parameters, since they are important in assessing maternal and fetal health, especially in normal pregnancy and preeclampsia.

Arterial blood gas and electrolyte analysis

The umbilical arterial blood was assayed immediately after collection for pH, blood gases, and co-oximetry on the fully automated analyzer Roche Cobas b 123 POC System. Blood collection was performed directly into a syringe coated as an anticoagulant with dry, balanced heparin, which prevented clotting and thus ensured homogeneity in the sample. Blood samples were processed within 15 minutes after collection to ensure optimal data integrity. If this was not possible, samples were stored on-ice and processed within 30 minutes after extraction was essential for the integrity of data. Blood gases and acid-base parameters, including pH, partial pressures of carbon dioxide (pCO2) and oxygen (pO2), bicarbonate (HCO3), base excess, and oxygen saturation (SpO2), oxyhemoglobin (O2Hb), carboxyhemoglobin (COHb), methemoglobin (MetHb), total bicarbonate (cHCO3), and standard bicarbonate (cHCO3-st) were measured. Electrolytes such as sodium (Na+), potassium (K+), chloride (Cl-), and calcium (Ca+) were also assessed.

CBC analysis

CBC testing was performed on the umbilical cord blood samples using a fully automated hematology analyzer (Sysmex XN-330), evaluating parameters including hemoglobin concentration, red and white blood cell counts, platelet count, and red cell indices including hematocrit value, mean corpuscular volume (MCV), mean corpuscular hemoglobin (MCH), and mean corpuscular hemoglobin concentration (MCHC).

Hb Electrophoresis analysis

The Hb Electrophoresis (hemoglobinopathy evaluation) test for the detection and analysis of Hb variants, HbA, HbF, and HbA2 fractions. This was done by completing the analysis with the use of the fully automated instrument H100 Hemoglobin Analyzer, based on principles from High-Performance Liquid Chromatography.”

4. Results and Discussion needs to be reformatted and rewritten.

Author response: Thank you for your valuable feedback. We appreciate your suggestion regarding the Results and Discussion section. We have reformatted and rewritten these sections to improve clarity, coherence, and alignment with journal standards.

We hope these changes meet your expectations and enhance the overall quality of the manuscript.

Please check “Revised Manuscript with Track Changes” file.

Reviewers' comments:

Reviewer #1: There are many sentences that do not correspond to academic and formal English writing. The quality of presentation needs to be improved.

For example, the following sentence looks good at first reading, but there is an editing error when considered in detail:

Analysis of demographic data comparing preeclampsia women and the control group displayed several key differences.

Instead of calling one group the women and the other the control group, a simpler fade and a sentence like Analysis of demographic data of the study groups (or of women with or without preeclampsia) revealed some differences may suffice.

Author response: As mentioned to the editor, we have carefully revised the manuscript for grammar, clarity, and overall flow. A professional language editing service was also consulted to ensure that the manuscript now meets academic and journal standards for English language quality. All corrections are highlighted in the 'Revised Manuscript with Track Changes' file.

Reviewer #2: The authors examined whether preeclampsia alters cord blood haematology and biochemistry. The manuscript shows that adult haemoglobin is reduced and foetal haemoglobin is increased in the cord blood of preeclampsia. I have some comments on the manuscript below:

Abstract:

Results, Lines 6-7 - Please remove ‘with lower platelet counts’ from the following statement as the direction of change has already been mentioned: ‘However, CBC results indicated lower platelets count in the cord blood of preeclamptic group, with lower platelet counts’.

Author response: The correction is done and ‘with lower platelet counts’ is removed

Conclusions, Lines 3-5 - The authors state that ‘…highlight the potential for using haemoglobin variants and CBC indices as biomarkers for early detection and management of the condition’. Would this be possible in routine clinical practice as the haemoglobin variants measured were in foetal blood and not maternal blood.

Author response: We appreciate your valuable feedback; we have mentioned this point in the Abstract section as follows (paragraph is highlighted in RED in the manuscript)

“While our study reveals significant alterations in fetal hemoglobin variants and CBC indices in the cord blood of preeclamptic pregnancies, the clinical applicability of these markers for early detection is currently limited by the inaccessibility of fetal blood before delivery. Nevertheless, these findings offer important insights into the hematological changes linked to preeclampsia. Future studies should explore the potential of detecting similar alterations in maternal blood as a more feasible and non-invasive approach for early diagnosis and risk assessment of preeclampsia.”

We also mentioned this point in our conclusion as follows (paragraph is highlighted in RED in the manuscript)

“ Although our study demonstrates significant alterations in fetal hemoglobin variants (HbF and HbA) and CBC indices in the cord blood of preeclamptic pregnancies, we acknowledge that the direct use of these fetal markers for early detection in routine clinical practice may be limited due to the inaccessibility of fetal blood prior to delivery. However, these findings may provide valuable pathophysiological insights into the hematological changes associated with preeclampsia. Future research should investigate whether similar alterations can be detected in maternal blood, which is more readily accessible during pregnancy. Identifying maternal biomarkers that reflect fetal or placental stress, such as free HbF or shifts in platelet i

---

## [Editor Report · Decision Letter 1]

25 Apr 2025

Hematological and Biochemical Alterations in Preeclampsia: Readings from Cord Blood Analysis

PONE-D-25-07466R1

Dear Dr. Siniyeh,

We’re pleased to inform you that your manuscript has been judged scientifically suitable for publication and will be formally accepted for publication once it meets all outstanding technical requirements.

Kind regards,

Apeksha Niraula, M.D.

Academic Editor

PLOS ONE
---

## [Editor Report · Acceptance letter]

PONE-D-25-07466R1

PLOS ONE

Dear Dr. Abu Siniyeh,

I'm pleased to inform you that your manuscript has been deemed suitable for publication in PLOS ONE. Congratulations! Your manuscript is now being handed over to our production team.

Kind regards,

on behalf of

Dr. Apeksha Niraula

Academic Editor

PLOS ONE